# THE GEOMETRY OF CONCEPTS:
# SPARSE AUTOENCODER FEATURE STRUCTURE

## ABSTRACT

Sparse autoencoders have recently produced dictionaries of high-dimensional vectors corresponding to the universe of concepts represented by large language models. We find that this concept universe has interesting structure at three levels: 1) The "atomic" small-scale structure contains "crystals" whose faces are parallelograms or trapezoids, generalizing well-known examples such as *(man:woman::king:queen)*. We find that the quality of such parallelograms and associated function vectors improves greatly when projecting out global distractor directions such as word length, which is efficiently done with linear discriminant analysis. 2) The "brain" intermediate-scale structure has significant spatial modularity; for example, math and code features form a "lobe" akin to functional lobes seen in neural fMRI images. We quantify the spatial locality of these lobes with multiple metrics and find that clusters of co-occurring features, at coarse enough scale, also cluster together spatially far more than one would expect if feature geometry were random. 3) The "galaxy" scale large-scale structure of the feature point cloud is not isotropic, but instead has a power law of eigenvalues with steepest slope in middle layers. We also quantify how the clustering entropy depends on the layer.

## 1    INTRODUCTION

The past year has seen a breakthrough in understanding how large language models work: sparse autoencoders have discovered large numbers of points ("features") in their activation space that can be interpreted as concepts (Huben et al., 2023; Bricken, 2024). Such SAE point clouds have recently been made publicly available (Lieberum et al., 2024), so it is timely to study their structure at various scales. This is the goal of the present paper, focusing on three separate spatial scales. In Section 3, we investigate if the "atomic" small-scale structure contains "crystals" whose faces are parallelograms or trapezoids, generalizing well-known examples such as *(man:woman::king:queen)*. In Section 4, we test if the "brain" intermediate-scale structure has functional modularity akin to biological brains. In Section 5, we study the "galaxy" scale large-scale structure of the feature point cloud, testing whether it is more interestingly shaped and clustered than an isosropic Gaussian distribution.

## 2    RELATED WORK

**SAE feature structure**: Sparse autoencoders have relatively recently garned attention as an approach for discovering interpretable language model features without supervision, with relatively few works examining SAE feature structure. Bricken et al. (2023) and Templeton et al. (2024) both visualized SAE features with UMAP projections and noticed that features tend to group together in "neighborhoods" of related features, in contrast to the approximately-orthogonal geometry observed in the toy model of Elhage et al. (2022). Engels et al. (2024) find examples of SAE structure where multiple SAE features appear to reconstruct a multi-dimensional feature with interesting geometry, and multiple authors have recently speculated that SAE vectors might contain more important structures (Mendel, 2024; Smith, 2024). Bussmann et al. (2024) suggest that SAE features are in fact linear combinations of more atomic features, and discover these more atomic latents with "meta SAEs". Our discussion of crystal structure in SAE features is related to this idea.

**Function vectors and Word embedding models**: Early word embedding methods such as GloVe and Word2vec, were found to contain directions encoding semantic concepts, e.g. the well-known formula f(king) - f(man) + f(woman) = f(queen) (Drozd et al., 2016; Pennington et al., 2014; Ma & Zhang, 2015). More recent research has found similar evidence of linear representations in sequence models trained only on next token prediction, including Othello board positions (Nanda et al., 2023; Li et al., 2022), the truth value of assertions (Marks & Tegmark, 2023), and numeric quantities such as longitude, latitude, birth year, and death year (Gurnee & Tegmark, 2023; Heinzerling & Inui, 2024). Recent works have found causal function vectors for in-context learning (Todd et al., 2023; Hendel et al., 2023; Kharlapenko et al., 2024), which induce the model to perform a certain task. Our discussion of crystal structures builds upon these previous works of finding task vectors or parallelogram structures in language models.

## 3 "ATOM" SCALE: CRYSTAL STRUCTURE

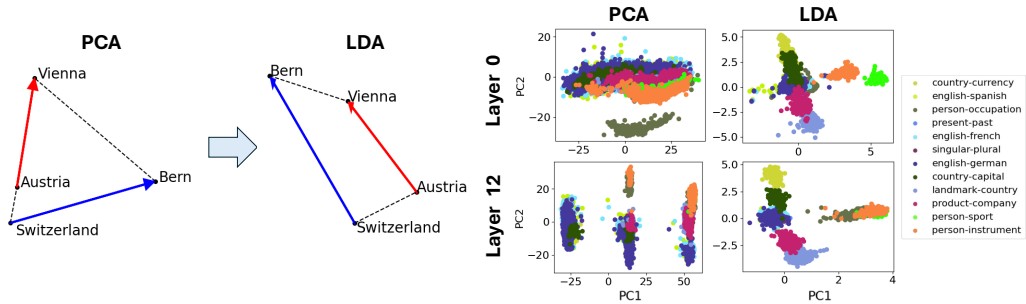

Figure 1: Parallelogram and trapezoid structure is revealed (left) when using LDA to project out distractor dimensions, tightening up clusters of pairwise Gemma-2-2b activation differences (right).

In this section, we search for what we term *crystal* structure in the point cloud of SAE features. By this we mean geometric structure reflecting semantic relations between concepts, generalizing the classic example of $(\mathbf{a}, \mathbf{b}, \mathbf{c}, \mathbf{d})$=(man,woman,king,queen) forming an approximate *parallelogram* where $\mathbf{b} - \mathbf{a} \approx \mathbf{d} - \mathbf{c}$. This can be interpreted in terms of two *function vectors* $\mathbf{b} - \mathbf{a}$ and $\mathbf{c} - \mathbf{a}$ that turn male entities female and turn entities royal, respectively. We also search for *trapezoids* with only one pair of parallel edges $\mathbf{b} - \mathbf{a} \propto \mathbf{d} - \mathbf{c}$ (corresponding to only one function vector); Fig. 1 (right) shows such an example with $(\mathbf{a}, \mathbf{b}, \mathbf{c}, \mathbf{d})$=(Austria,Vienna,Switzerland,Bern), where the function vector can be interpreted as mapping countries to their capitals.

We search for crystals by computing all pairwise difference vectors and clustering them, which should result in a cluster corresponding to each function vector. Any pair of difference vectors in a cluster should form a trapezoid or parallelogram, depending on whether the difference vectors are normalized or not before clustering (or, equivalently, whether we quantify similarity between two difference vectors via Euclidean distance or cosine similarity).

Our initial search for SAE crystals found mostly noise. To investigate why, we focused our attention on Layers 0 (the token embedding) and 1, where many SAE features correspond to single words. We then studied Gemma2-2b residual stream activations for previously reported word $\mapsto$ word function vectors from the dataset of (Todd et al., 2023), which clarified the problem. Figure 1 illustrates that candidate crystal quadruplets are typically far from being parallelograms or trapezoids. This is consistent with multiple papers pointing out that (man,woman,king,queen) is not an accurate parallelogram either.

We found the reason to be the presence of what we term *distractor features*. For example, we find that the horizontal axis in Figure 1 (right) corresponds mainly to word length (Appendix B, Figure 10), which is semantically irrelevant and wreaks havoc on the trapezoid (left), since "Switzerland" is much longer than the other words.

To eliminate such semantically irrelevant distractor vectors, we wish to project the data onto a lower-dimensional subspace orthogonal to them. For the (Todd et al., 2023) dataset, we do this with

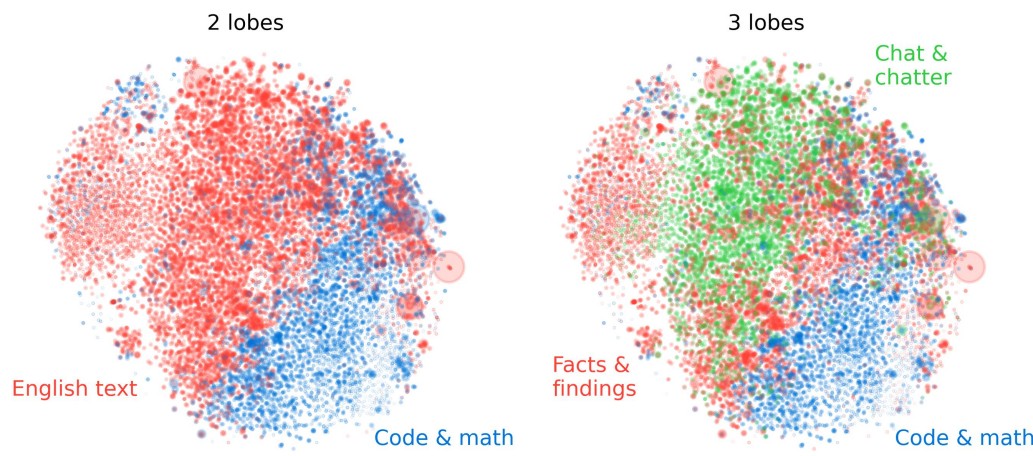

Figure 2: Features in the SAE point cloud identified that tend to fire together within documents are seen to also be geometrically co-located in functional "lobes", here down-projected to 2D with t-SNE with point size proportional to feature frequency. A 2-lobe partition (left) is seen to the point cloud into roughly equal parts, active on code/math documents and English language documents, respectively. A 3-lobe partition (right) is seen to mainly subdivide the English lobe into a part for short messages and dialogue (e.g. chat rooms and parliament proceedings) and one primarily containing long-form scientific papers.

Linear Discriminant Analysis (LDA) (Xanthopoulos et al., 2013), which projects onto signal-to-noise eigenmodes where "signal" and "noise" are defined as the covariance matrices of inter-cluster variation and intra-cluster variation, respectively. Figure 1 illustrates that this dramatically improves the cluster and trapezoid/parallelogram quality, highlighting that distractor features can hide existing crystals.

## 4 "BRAIN" SCALE: MESO-SCALE MODULAR STRUCTURE

We now zoom out and look for larger-scale structure. In particular, we investigate if *functionally* similar groups of SAE features (which tend to fire together) are also *geometrically* similar, forming "lobes" in the activation space.

In animal brains, such functional groups are well-known clusters in the 3D space where neurons are located. For example, Broca's area is involved in speech production, the auditory cortex processes sound, and the amygdala is primarily associated with processing emotions. We are curious whether we can find analogous functional modularity in the SAE feature space.

We test a variety of methods for automatically discovering such functional "lobes" and for quantifying if they are spatially modular. We define a **lobe partition** as a partition of the point cloud into $k$ subsets ("lobes") that are computed *without positional information*. Instead, we identify such lobes based on then being *functionally* related, specifically, tending to fire together within a document.

To automatically identify functional lobes, we first compute a histogram of SAE feature co-occurrences. We take gemma-2-2b and pass documents from The Pile Gao et al. (2020) through it. At a given layer, we record the SAE features that fire (for the GemmaScope JumpReLU SAEs, we count a feature as firing if its hidden activation $> 1$). Features are counted as co-occurring if they both fire within the same block of 256 tokens – this length provides a coarse "time resolution" allowing us to find tokens that tend to fire together within the same document rather than just at the same token. We use a max context length of 1024, and only use one such context per document, giving us at most 4 blocks (and histogram updates) per document of The Pile. We compute histograms across 50k documents. Given this histogram, we compute an affinity score between each pair of SAE features based on their co-occurrence statistics and perform spectral clustering on the resulting affinity matrix.

We experiment with the following notions of co-occurrence-based affinity: simple matching coefficient, Jaccard similarity, Dice coefficient, overlap coefficient, and Phi coefficient, which can all be computed just from a co-occurrence histogram. In the Appendix A.1, we review definitions for each of these and illustrate how the choice between them affects the resulting lobes.

Our null hypothesis is that functionally similar points (of commonly co-occurring SAE features) are uniformly distributed throughout the activation space, showing no spatial modularity. In contrast, Figure 2 shows lobes that appear visually quite spatially localized. To quantify how statistically significant this is, we use two approaches to rule out the null hypothesis.

1. While we can cluster features based on whether they co-occur, we can also perform spectral clustering based on the cosine similaity between SAE feature decoder vectors. Given a clustering of SAE features using cosine similarity and a clustering using co-occurrence, we compute the mutual information between these two sets of labels. In some sense, this directly measures the amount of information about geometric structure that one gets from knowing functional structure. We report the adjusted mutual information Vinh et al. (2009) as implemented by scikit-learn Pedregosa et al. (2011), which corrects for chance agreements between the clusters.

2. Another conceptually simple approach is to train models to predict which functional lobe a feature is in from its geometry. To do this, we take a given set of lobe labels from our co-occurrence-based clustering, and train a logistic regression model to predict these labels directly from the point positions, using an 80-20 train-test split and reporting the balanced test accuracy of this classifier.

Figure 4 shows that for both measures, the Phi coefficient wins, delivering the best correspondence between functional lobes and feature geometry. To show that this is statistically significant, we randomly permute the cluster labels from the cosine similarity-based clustering and measure the adjusted mutual information. We also randomly re-initialize the SAE feature decoder directions from a random Gaussian and normalize, and then train logistic regression models to predict functional lobe from these feature directions. Figure 4 (bottom) shows that both tests rule out the null hypothesis at high significance, at 954 and 74 standard deviations, respectively), clearly demonstrating that the lobes we see are real and not a statistical fluke.

To assess what each lobe specializes in, we run 10k documents from The Pile through gemma-2-2b, and run an SAE on the model's activations at the layer 12 residual stream. Within blocks of 256-tokens, we record the SAE features which fire. We then record which lobe has the highest proportion of its features firing on that block of tokens. Each document in The Pile is attached with a name specifying the subset of the corpus that document is from. For each document type, for each 256-token block within a document of that type, we record which lobe which had the highest proportion of its SAE features firing. Across thousands of documents, we can then look at a histogram of which lobes were maximally activating across each document type. We show these results for three lobes, computed with the phi coefficient as the co-occurrence measure, in Figure 3. This forms the basis for our lobe labeling in Figure 2.

The effects of the five different co-occurrence measures are compared in Fig. 5. Although we found Phi to be best, all five are seen to discover the "code/math lobe".

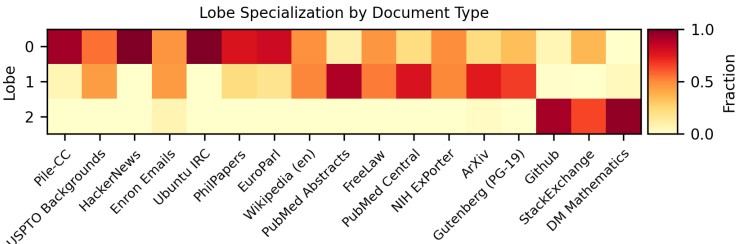

Figure 3: Fraction of contexts in which each lobe had the highest proportion of activating features. For each document type, these fractions sum to 1 across the lobes. We see that lobe 2 typically disproportionately activates on code and math documents. Lobe 0 and 1 activate on other documents, with lobe 0 activating more on documents containing short text and dialogue (chat comments, parliamentary proceedings) and lobe 1 activating more on scientific papers.

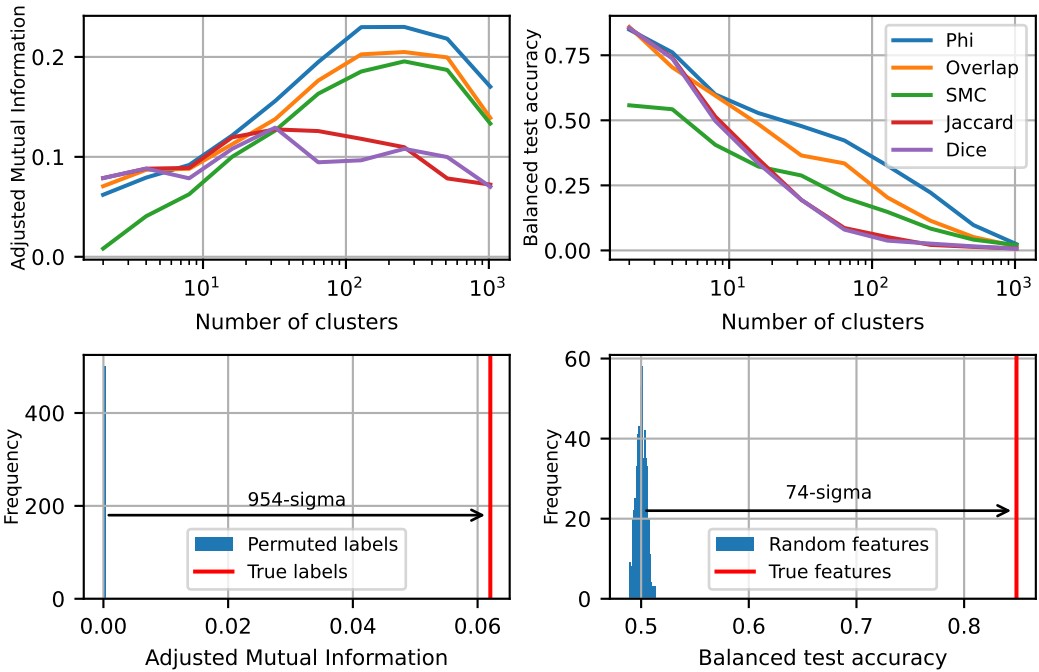

Figure 4: Top Left: Adjusted mutual information between spatial clusters and functional (cooccurrence-based) clusters. Top Right: logistic regression balanced test accuracy, predicting cooccurrence-based cluster label from position. Bottom Left: Adjusted mutual information with randomly-permuted cosine similarity-based clustering labels. Bottom Right: balanced test accuracy with random unit-norm feature vectors. The statistical significance reported is for Phi-based clustering with into lobes.

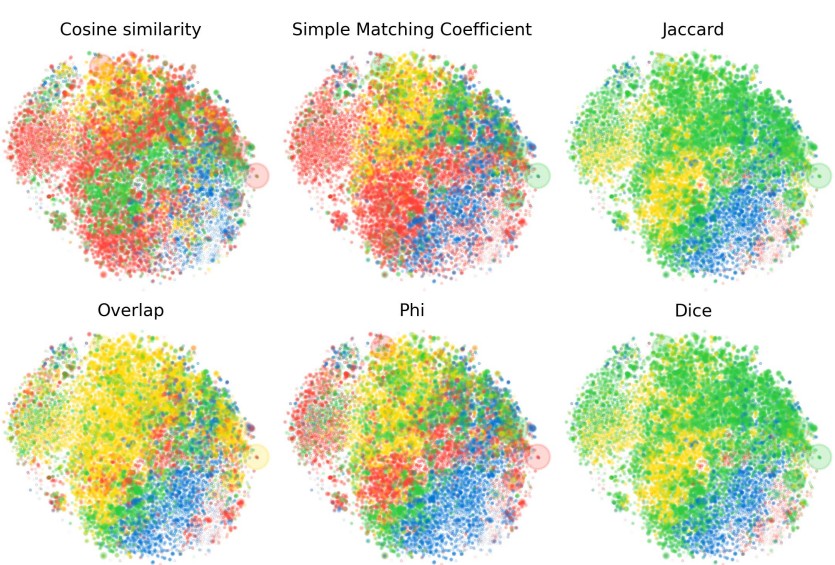

Figure 5: Comparison of the lobe partitions of the SAE point cloud discovered with different affinity measures, with the same t-SNE projection as Figure 2. In the top left, we show clusters computed from **geometry**, the angular similarity between features as the affinity score for spectral clustering. All other measures are based on whether SAE features co-occur (fire together) within 256-token blocks, using different measures of affinity. Although Phi predicts spatial structure best, all co-occurrence measures are seen to discover the "code/math lobe".

## 5 "GALAXY" SCALE: LARGE-SCALE POINT CLOUD STRUCTURE

In this section, we zoom out further and study the "galaxy" scale structure of the point cloud, mainly its overall shape and clustering, analogously to how astronomers study galaxy shapes and substructure.

The simple null hypothesis that we try to rule out is that the point cloud is simply drawn from an isotropic multivariate Gaussian distribution. Figure 6 visually suggests that the cloud is not quite round even in its three first principal components, with some principal axes slightly wider than others akin to a human brain.

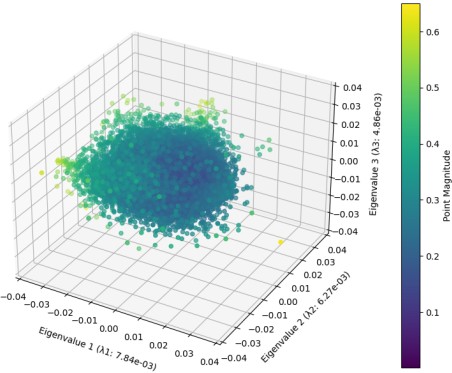

Figure 6: 3D Point Cloud visualizations of top PCA components for the Gemma2-2b layer 12 SAE features.

### 5.1 SHAPE ANALYSIS

Figure 6 (left) quantifies this by showing the eigenvalues of the point cloud's covariance matrix in decreasing order. revealing that they are not constant, but appear to fall off according to a power law. To test whether this surprising power law is significant, the figure compares it with the corresponding eigenvalue spectrum for a point cloud drawn from an isotropic Gaussian distribution, which is seen to be much flatter and consistent with the analytic prediction: the covariance matrix of $N$ random vectors from a multivariate Gaussian distribution follow a Wishart Distribution, which is well studied in random matrix theory. Since the abrupt dropoff seen for the smallest eigenvalues is caused by limited data and vanishes in the limit $N \to \infty$, we dimensionally reduce the point clound to its 100 largest principal componends for all subsequent analysis in this section. In other words, the point cloud has the shape of a "fractal cucumber", whose width in successive dimensions falls off like a power law. We find such power law scaling is significantly less prominent for activations than for SAE features; it will be interesting for further work to investigate its origins.

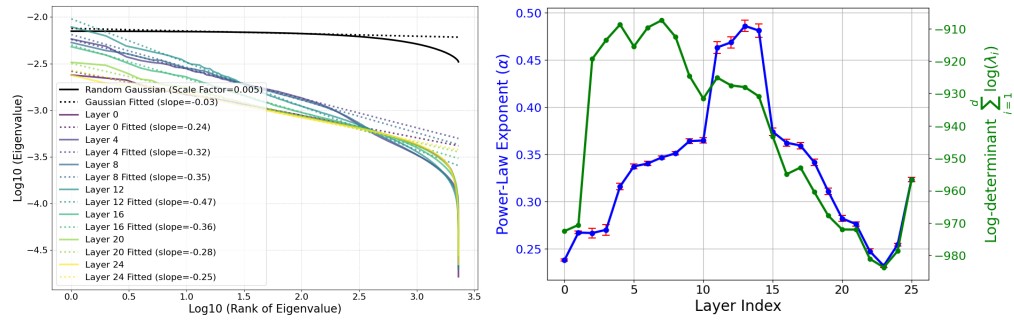

Figure 7: Eigenvalues of the point cloud are seen to decay as an an approximate power law (left), whose slope depends on layer (right) and is strongly inconsistent with sampling from an isotropic Gaussian distribution. The effective point cloud volume is also seen to be layer-dependent (right).

Figure 6 (right) shows how the slope of the aforementioned power law depends on LLM layer, computed via linear regression against the 100 largest eigenvalues. We see a clear pattern where middle layers have the steepest power law slopes: (Layer 12 has slope -0.47, while early and late layers (e.g., Layers 0 and 24) have shallower slopes (-0.24 and -0.25), respectively. This may hint that middle layers act as a bottleneck, compressing information into fewer principal components, perhaps optimizing for more efficient representation of high-level abstractions. Figure 7 (right) also shows how effective cloud volume (the determinant of the covariance matrix) depends on layer, on a logarithmic scale.

## 5.2 Clustering analysis

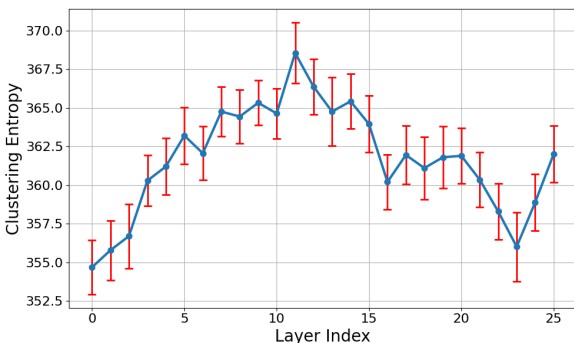

Figure 8: Estimated clustering entropy across layers with 95% confidence intervals. Middle layers exhibit reduced clustering entropy, while earlier and later layers show higher entropy, indicating distributed and concentrated feature representation in early and late layers, respectively.

Clustering of galaxies or microscopic particles is often quantified in terms of a power spectrum or correlation function. This is complicated for our very high-dimensional data, since the underlying density of varies with radius and, for a high-dimensional Gaussian distribution, is strongly concentrated around a relatively thin spherical shell. For this reason, we instand quantify clustering by estimating the *entropy* of the distribution that the point cloud is assumed to be sampled from. We estimate the entropy $H$ from our SAE feature point cloud using on the $k$-th nearest neighbor (k-NN) method Dasarathy (1991); Kozachenko & Leonenko (1987), computed as follows,

$$H_{features} = \frac{d}{n} \sum_{i=1}^{n} \log(r_i + \theta) + \log(n-1) - \Psi \tag{1}$$

where $r_i$ is the distance to the $k$-th nearest neighbor for point $i$, and $d$ is the dimensionality of the point cloud; $n$ is the number of points; the constant $\Psi$ is the digamma term from the k-NN estimation. As a baseline, the Gaussian entropy represents the maximum possible entropy for a given covariance matrix. For a Gaussian distribution with the same covariance matrix, the entropy computed as:

$$H_{gauss} = \frac{d}{2}\big(1 + \log(2\pi)\big) + \sum_{i=1}^{d} \log(\lambda_i) \tag{2}$$

where $\lambda_i$ are the eigenvalues of the covariance matrix. We define the **clustering entropy** (often referred to as "negentropy" in physics as $H_{gauss} - H$, i.e., how much lower the entropy is than its maximum allowed value. Figure 8 shows the estimated clustering entropy across different layers. We see that the SAE point cloud is strongly clustered, particulary in the middle layers. In future work, it will be interesting to examine whether these variations depend mainly on the prominence of crystals or lobes in different layers, or have an altogether different origin.

## 6 CONCLUSION

In this paper, we have found that the concept universe of SAE point clouds has interesting structures at three levels: 1) The "atomic" small-scale structure contains "crystals" whose faces are parallelograms or trapezoids, generalizing well-known examples such as *(man:woman::king:queen)*, and get revealed when projecting out semantically irrelevant distractor features. . 2) The "brain" intermediate-scale structure has significant spatial modularity; for example, math and code features form a "lobe" akin to functional lobes seen in neural fMRI images. 3) The "galaxy" scale large-scale structure of the feature point cloud is not isotropic, but instead has a power law of eigenvalues with steepest slope in middle layers. We hope that our findings serve as a stepping stone toward deeper understanding of SAE features and the workings of large language models.

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

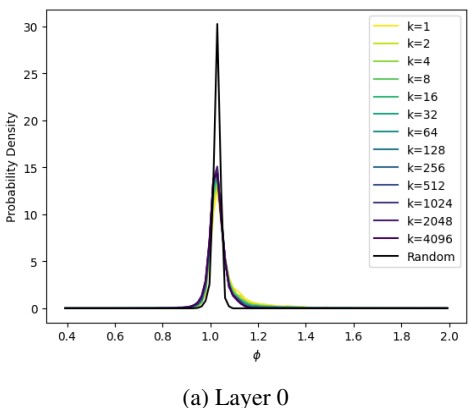 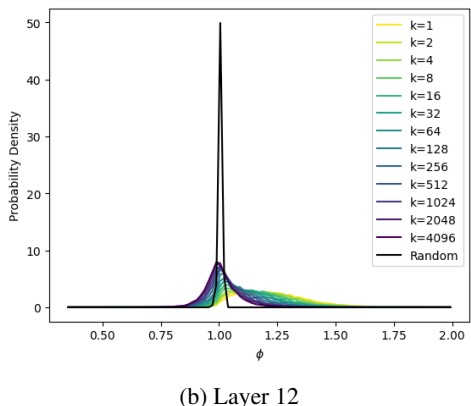

| (a) Layer 0 | (b) Layer 12 |

Figure 9: Histogram, over all features, of Phi coefficient with $k$-th nearest cosine similarity neighbor. Each line represents a different $k$. The "random" line is plotted by drawing a random feature for each feature, then computing the Phi coefficient. Features with higher cosine similarity have higher Phi coefficient, but this is less pronounced in layer 0 compared to layer 12.

Petros Xanthopoulos, Panos M Pardalos, Theodore B Trafalis, Petros Xanthopoulos, Panos M Pardalos, and Theodore B Trafalis. Linear discriminant analysis. *Robust data mining*, pp. 27–33, 2013.

G. Udny Yule. On the methods of measuring association between two attributes. *Journal of the Royal Statistical Society*, 75(6):579–652, 1912.

## A ADDITIONAL INFORMATION ON BRAIN LOBES

### A.1 CO-OCCURRENCE MEASURES

Definitions of co-occurrence based affinity measures: Let $n_{ij}$ be the number of times features $i$ and $j$ co-occur. Let $m_{11}$ be number of times $i$ and $j$ co-occur, $m_{00}$ be number of times $i$ and $j$ both do not occur, $m_{10}$ be number of times $i$ occurs but $j$ does not, $m_{1\bullet}$ be number of times $i$ occurs and $j$ either occurs or not, and so on. Then,

Jaccard similarity, Jaccard (1908):

$$J_{ij} = \frac{|i \cap j|}{|i \cup j|} = \frac{n_{ij}}{n_{ii} + n_{jj} - n_{ij}}$$

Dice score, Dice (1945):

$$DSC_{ij} = \frac{2|i \cap j|}{|i| + |j|} = \frac{2n_{ij}}{n_{ii} + n_{jj}}$$

Overlap coefficient:

$$overlap_{ij} = \frac{|i \cap j|}{\min(|i|, |j|)} = \frac{n_{ij}}{\min(n_{ii}, n_{jj})}$$

Simple matching coefficient:

$$SMC_{ij} = \frac{m_{00} + m_{11}}{m_{00} + m_{11} + m_{01} + m_{10}}$$

Phi coefficient, Yule (1912):

$$\phi_{ij} = \frac{m_{11}m_{00} - m_{10}m_{01}}{\sqrt{m_{1\bullet}m_{0\bullet}m_{\bullet1}m_{\bullet0}}}$$

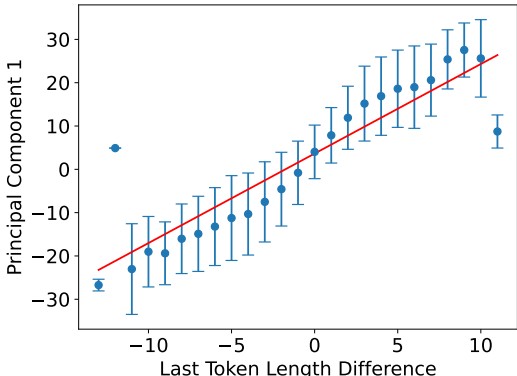

Figure 10: Plot of the first principal component in the difference space as a function of last token length difference in Gemma-2-2b layer 0. The linear relationship indicates that the first principal component encodes the length difference between two words' last tokens.

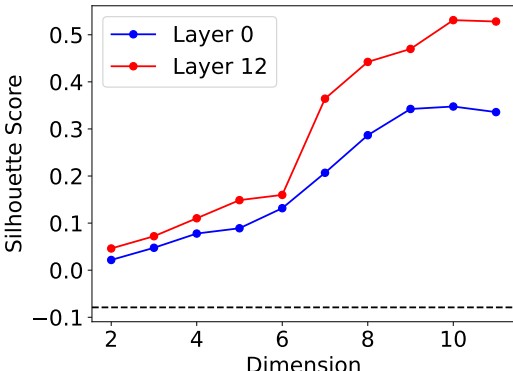

Figure 11: Silhouette Score, a measure of clustering quality, as a function of reduced dimension in LDA. The plot indicates that training an affine transformation for semantic cluster separation is easier in mid-layer (layer 12), where the model starts to develop concept-level understanding of the input.

## B  UNDERSTANDING PRINCIPAL COMPONENTS IN DIFFERENCE SPACE

Figure 10 shows that the first principal component encodes mainly the length difference between two words' last tokens in Gemma-2-2b Layer 0.

## C  BREAKING DOWN SAE VECTORS BY PCA COMPONENT

An additional investigation of structure we undertake is quantifying how SAE vectors are distributed throughout the PCA components of the activations vectors. To do this, we define a PCA score:

$$\texttt{PCA score}(feature_j) = \frac{1}{n} \sum_i i * (pca_i @ feature_j)^2$$

This metric is a weighted sum between $0$ and $1$ measuring approximately where in the PCA each SAE feature lies. In Figure 12, we plot this metric on a single Gemma Scope SAE (the results look similar on all Gemma Scope SAEs), and we see that there is an intriguing dip into earlier PCA features in the last third of SAE features.

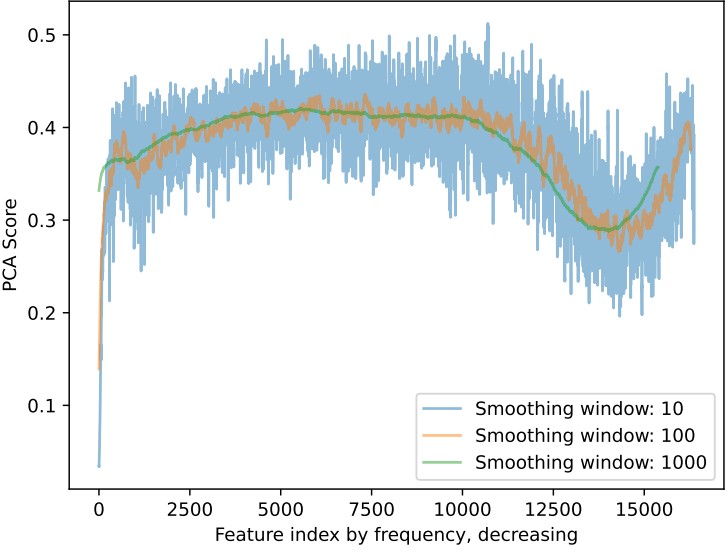

Figure 12: Smoothed PCA scores for each SAE feature of the layer 12, width $16k$, $L_0 = 176$ Gemma Scope 2b SAE, sorted by frequency. PCA score $= \frac{1}{n} \sum_i i * (pca_i @ feature_j)^2$, where $n$ is the number of PCA features. The smoothed curves just average this somewhat noisy metric over adjacent sorted features. This measures approximately where in the PCA each SAE feature lies, and shows that there is a dip into earlier PCA features in the last third of SAE features.

