# OpenReview forum: "The Geometry of Concepts: Sparse Autoencoder Feature Structure"
_ICLR.cc/2025/Conference — ICLR 2025 Conference Withdrawn Submission_

### Official Review · Reviewer_s5fj · 2024-10-30

**Soundness:** 1
**Presentation:** 1
**Contribution:** 1
**Rating:** 1
**Confidence:** 5

**Summary:**

This paper analyzes sparse autoencoders and provides an interpretation of the structure that they learn. They provide analogies to the low, mid and high level in terms of solid state physics, neuroscience, and astronomy respectively.

**Strengths:**

I believe these comparisons are certainly original and that this paper has the potential to help understand a specific but common architecture for deep learning. Some of the figures look nice.

**Weaknesses:**

This should have been submitted in the next cycle rather than rushed for this one. This was not reviewed by the authors for clarity, with issues ranging from minor (first entry of bibliogrophy an erroneous duplicate of the second) to the critical (the introduction + related work combined are 3 paragraphs). These sections are so spartan that I am unable ascertain what the authors were necessarily trying to accomplish, much less whether they succeeded in their goal. The conclusion does not enlighten at all.

**Questions:**

None

---

### Official Review · Reviewer_oYQC · 2024-10-30

**Soundness:** 1
**Presentation:** 2
**Contribution:** 2
**Rating:** 3
**Confidence:** 3

**Summary:**

This paper examines multi-scale structures in sparse autoencoder (SAE) feature spaces within language models. It proposes that SAE features exhibit patterns at three levels: atomic (e.g., crystal shapes like parallelograms), brain-scale (modularity similar to brain lobes), and galaxy-scale (clustering with a power-law eigenvalue distribution). The study applies clustering, dimensionality reduction, and co-occurrence measures to suggest a structured feature space, potentially advancing model interpretability.

**Strengths:**

- The multi-scale structural analysis is ambitious, with an interesting attempt to parallel functional modularity in neural networks to concepts from biological brains.
- The methodological variety is a strength, as the paper employs clustering and linear discriminant analysis to explore SAE feature structure at multiple levels.
- Visualizations of proposed structures (e.g., crystal formations and modular lobes) aid in explaining complex ideas, though they lack robust validation.

**Weaknesses:**

- The writing is inconsistent, with rough sections, minor errors (e.g., “Instand” on line 407), and inconsistent citation formats, suggesting a need for refinement. Additionally, the paper’s presentation lacks depth in essential sections, more closely resembling an extended abstract than a full conference paper.
- The paper does not make full use of the standard length for conference papers, leaving room to expand the introduction, clarify methodology, and provide further experimental detail. For example, the introduction lacks comprehensive background or positioning within existing literature, and the methodology could be detailed to improve clarity. Additionally, there is no supplementary code provided, which limits reproducibility and replicability.
- The identified patterns and modular "lobes" have limited novelty, as these structural insights are well-documented in language model literature. The study does not significantly advance the current understanding of SAE feature structures or provide actionable interpretability benefits.
- Claims of crystal structures and functional lobes are interesting but lack rigorous validation. The parallelogram and trapezoid analogies remain speculative without robust quantitative analysis, making it unclear whether these patterns consistently represent SAE structures or are isolated findings.
- The clustering methods are not sufficiently justified or compared with alternatives that might yield more interpretable groupings, such as unsupervised methods better suited to high-dimensional data.

**Questions:**

See above sections for details.

---

### Official Review · Reviewer_Vkqp · 2024-11-03

**Soundness:** 3
**Presentation:** 2
**Contribution:** 2
**Rating:** 3
**Confidence:** 2

**Summary:**

This paper explores the geometry of structures in the relationship between concepts as revealed by sparse autoencoder in LLM at different scales: "crystals" at the atomic scale, spatially localized modules at the brain scale, and anisotropy of the feature point clouds in the so-called galaxy scale.   Factorizing out the distractor dimensions using LDA has been shown to be helpful in revealing concept clusters better. The co-occurring statistics between pairs of SAE features reveal spatialized localized functional modules. On a large scale, the SAE point cloud is found to be strongly clustered and follows power laws.

**Strengths:**

The paper systematically studied some geometric properties in the relationship among concepts in LLM model at different scales, and made some interesting observations.

**Weaknesses:**

However, the computational and functional implications of these observations on the geometry of concepts relationship at different scales need to be clarified.   Why are these observations interesting? Why do they matter? What are their utilities? How do they help us understand or improve LLM? It would be helpful to address these basic questions in the introduction and discussion.

**Questions:**

Why are these observations interesting? Why do they matter? What are their utilities? How do they help us understand or improve LLM? It would be helpful to address these basic questions in the introduction and discussion.

---

### Official Review · Reviewer_pmQU · 2024-11-03

**Soundness:** 2
**Presentation:** 2
**Contribution:** 2
**Rating:** 3
**Confidence:** 3

**Summary:**

The contribution concerns the structure of latent spaces in LLMs seen through the lense of large sparse autoencoders. This is a field of considerable interest and here interesting hypotheses are explore, including analysis at multiple scales from local to global structure.

All analyses are empirical and based on the GemmaScope open source SAEs.

 At the local scale generalized functional relations are found with an interesting de-confounder projection to discover relations as "king-man"="queen-woman" ="crowning”). At the intermediate scale are found distributed areas corresponding to topics, while interesting asymmetries are found at the global scale.

**Strengths:**

Several contributions are presented including projecting confounders away at the local scale to recover functional descriptions. At the intermediate level t-SNE plots are used to develop topical distributions. At the global scale a null hypothesis of isotropic covariance is rejected, to discover unexpected structure in the eigenvalues (viz the fractal cucumber mentioned in L361)

**Weaknesses:**

While intriguing, the whole set of analyses appear quite basic and findings appear anecdotal. Unclear how the different dimensions of the Gemma Scope data enter the analyses.

 All in all the investigations could be better motivated. Would be interesting to link the discussion with human alignment and known representational relations in human culture.

The terms used to denote functional scales: "atom", "brain" and "galaxy", could be better motivated.

**Questions:**

Make the novelty more clear - e.g., relative to already published work on SAEs and Gemma Scope in particular (viz. the work using UMAP rather than t-SNE for visualization of topics).

Consider discussing the structure found in the context of the work reported in "related work"

Would there be a away to understand the similarities and differences in Fig 5?

How robust are the results in Fig 7 wrt different latent space dimensions of the SAE?

---

### Official Review · Reviewer_Yg4m · 2024-11-04

**Soundness:** 1
**Presentation:** 1
**Contribution:** 2
**Rating:** 3
**Confidence:** 3

**Summary:**

The manuscript claims to study the structure of features (point clouds) obtained using sparse autoencoders on some layers of the LLM Gemma. These features can be interpreted as concepts according to the literature cited. The structure is studied at three different scales, from fine to coarse: pairwise discriminability of concepts (with invariance to irrelevant distractors); clustering of concepts (comparing clustering by geometry alone and functional clustering); and global shape of the point cloud. Clustering and global shape are compared against simple null hypotheses (uniform clusters with no spatial relations, and isotropic gaussian blob, respectively).

**Strengths:**

Evocative language that promises (but then fails to) to explore an interesting connection across domains of knowledge.

The approach to compare the similarity of geometric and functional clusters is interesting.

There is value to performing a purely exploratory study of the feature space of a LLM, this might be an interesting starting point for the authors to identify a direction worth pursuing in a rigorous manner in the future.

**Weaknesses:**

The manuscript is written casually and in a rush. Methods are explained poorly. Evocative words from disparate fields are thrown together with no in-depth argument for why the analogies are meaningful.
Action: I suggest writing a Methods or Approach section, before presenting results and interpretations.

The manuscript fails to convey how novel and unexpected the findings are, and what is the contribution. In particular, are the uniform clustering and the spherical gaussian valid even as a strawman? Is there some advance with respect to other literature?

Just as the writing is casual (“wreaks havoc”, “fire together” used with no explanation and then defined two paragraphs later; figure 4 explained before figure 3 before figure 2, and figure 2 caption using concepts defined in fig 3 without a pointer; many typos), so is the analysis.
Action: figures should be ordered according to when they are referred to in the text. Nex concepts and jargon (e.g. 'fire together') should be defined on first use.

As stated above, I believe there’s value in exploratory analysis, but for a manuscript to have value to the readers perhaps there should be some in-depth analysis. But as an outsider to this field, I don’t know the standards and I might be unaware that a quick first-pass of the data is worth publishing.

I am not a chemist or cosmologist, so I have a hard time judging if the analogies to crystals and galaxies are more than a very loose analogy (the brain lobe analogy is quite superficial though: “some principal axes slightly wider than others akin to a human brain”… with literally no additional articulation of this particular proposal, a watermelon would be as good a metaphor). If one could actually learn something about LLMs from leveraging the maths and physics of crystals and galaxies, that would be valuable. But this manuscript is quite far from that, in applying basic first-pass data analytics tools that are not specific to any field.
Action: please provide more detailed explanations of how these analogies relate to the LLM features, or justify why these particular analogies were chosen over others.

**Questions:**

What are the significant results? What is novel?

Why are unform clustering and spherical gaussian meaningful/plausible alternative outcomes?
Action: please explain the rationale behind choosing these specific null hypotheses and how they relate to current understanding or assumptions about LLM feature spaces.

Why is the power law “surprising” in the shape analysis, in what way is it not expected?
Action: please provide context on what distribution or behavior they expected to see and why, which would help readers understand the significance of the power law finding.

The five co-occurrence measures “all discover the code math lobe”. What about the other clusters? Are the differences between measures informative? Or worrying?

---

### Note · Authors · 2024-11-26

**Comment:**

We'd like to thank the reviewers for their comments and suggestions. Clearly, we have some work to do on improving the presentation of the paper and on clarifying its contributions, and so we are withdrawing the paper from consideration at ICLR 2025.

**Withdrawal Confirmation:**

I have read and agree with the venue's withdrawal policy on behalf of myself and my co-authors.